# Spatial modelling for population replacement of mosquito vectors at continental scale

**Nicholas J. Beeton**[1]*, **Andrew Wilkins**[2]*, **Adrien Ickowicz**[1], **Keith R. Hayes**[1], **Geoffrey R. Hosack**[1]

1 Data61, CSIRO, 3 Castray Esplanade, Battery Point TAS, Australia, 2 Mineral Resources, CSIRO, 1 Technology Court, Pullenvale QLD, Australia

* nick.beeton@csiro.au (NJB); andy.wilkins@csiro.au (AW)

**Data Availability Statement:** All code is available here: https://doi.org/10.25919/rgqc-7520.

**Funding:** NJB, GRH, AI and KRH contribution to this study was funded by CSIRO Data61 and

## Abstract

Malaria is one of the deadliest vector-borne diseases in the world. Researchers are developing new genetic and conventional vector control strategies to attempt to limit its burden. Novel control strategies require detailed safety assessment to ensure responsible and successful deployments. *Anopheles gambiae* sensu stricto (s.s.) and *Anopheles coluzzii*, two closely related subspecies within the species complex *Anopheles gambiae* sensu lato (s.l.), are among the dominant malaria vectors in sub-Saharan Africa. These two subspecies readily hybridise and compete in the wild and are also known to have distinct niches, each with spatially and temporally varying carrying capacities driven by precipitation and land use factors.

We model the spread and persistence of a population-modifying gene drive system in these subspecies across sub-Saharan Africa by simulating introductions of genetically modified mosquitoes across the African mainland and its offshore islands. We explore transmission of the gene drive between the two subspecies that arise from different hybridisation mechanisms, the effects of both local dispersal and potential wind-aided migration to the spread, and the development of resistance to the gene drive. Given the best current available knowledge on the subspecies' life histories, we find that an introduced gene drive system with typical characteristics can plausibly spread from even distant offshore islands to the African mainland with the aid of wind-driven migration, with resistance beginning to take over within a decade. Our model accounts for regional to continental scale mechanisms, and demonstrates a range of realistic dynamics including the effect of prevailing wind on spread and spatio-temporally varying carrying capacities for subspecies. As a result, it is well-placed to answer future questions relating to mosquito gene drives as important life history parameters become better understood.

## Author summary

Conventional control methods have dramatically reduced malaria, but it still kills over 300,000 children in Africa each year, and this number could increase as their effectiveness wanes. Novel control methods using gene drives rapidly reduce or modify malaria vector

CSIRO Health and Biosecurity. AW contribution was funded by CSIRO Health and Biosecurity and CSIRO Mineral Resources. The CSIRO funders had no role in study design, data collection and analysis, decision to publish, or preparation of the manuscript.

**Competing interests:** The authors have declared that no competing interests exist.

populations in laboratory settings, and hence are now being considered for field applications. We use modelling to assess how a gene drive might spread and persist in the malaria-carrying subspecies *Anopheles gambiae* sensu stricto (s.s.) and *Anopheles coluzzii*. These two subspecies interbreed and compete, so we model how these interactions affect the spread of the drive at a continental scale. In scenarios that allow mosquitoes to travel on prevailing wind currents, we find that a gene drive can potentially spread across national borders—and jump from offshore islands to the African mainland—but spread is eventually arrested when the drive allele is ousted by a resistant allele. As we learn more about the population dynamics of both genetically modified and wild mosquitoes, and as gene drive systems are further developed to allow local containment and evade resistance, our model will be able to answer more detailed questions about how they can be applied in the field effectively and safely.

## Introduction

Contemporary malaria control interventions—insecticide treated bed nets, indoor residual spraying and artemisinin based combination therapy—have dramatically reduced the burden of malaria in Africa [1]. Since 2017, however, the rate of progress on malaria reduction has stalled and in 2019 malaria still claimed an estimated 389,000 African lives, mainly children under 5 years of age [2]. At least 99% of these cases are caused by *Plasmodium falciparum*, transmitted by a small number of dominant malaria vectors, most notably *Anopheles arabiensis*, *Anopheles coluzzii*, *Anopheles gambiae* sensu stricto (s.s.) and *Anopheles funestus* [3].

The ongoing burden of malaria, together with increasing rates of insecticide resistance in malarial vector mosquitoes [3], has motivated proposals to develop new genetic control strategies, including: a) self-limiting, population suppression methods that induce male sterility [4, 5] or male bias [6, 7]; b) self-sustaining (gene drive), suppression methods that induce female sterility [8, 9]; and, c) self-sustaining, population replacement methods that make vectors refractory for the malaria parasite [10, 11]. Any proposal to test these genetic control strategies outside of contained laboratory settings will likely require a detailed quantitative risk assessment that predicts the potential spread and persistence of transgenic mosquitoes from release sites, and the possible introgression of a transgenic construct into closely related species through interspecific mating [12, 13]. Spatial models of spread and persistence are also needed to describe the dynamics of important gene drive processes such as the development of resistance to the gene drive [14]. Quantitative spatial models have been developed for the spread and persistence of self-limiting, population suppressing constructs [7, 15], together with self-sustaining, population-modifying [16–19] and population-suppressing [20–22] constructs.

This paper models the spread and persistence of a population-modifying gene drive system [23, 24] in *Anopheles gambiae* s.s. and *Anopheles coluzzii* across sub-Saharan Africa. These two subspecies, which are often modelled as single group, are together with *Anopheles arabiensis* and *Anopheles funestus* the dominant vectors of malaria in sub-Saharan Africa. *An. gambiae* s.s. and *An. coluzzii* are both highly anthropophilic and efficient malaria vectors. The two subspecies are closely related enough to interbreed but hybridisation rates vary in space and time [25]. They also have different larval habitat preferences [26], and *An. coluzzii* is thought to have a superior resistance to desiccation stress [27], hence is more drought tolerant than *An. gambiae* s.s. [28]. Although they are defined as two separate species by [29], we refer to them here and subsequently as subspecies to emphasise the lack of reproductive isolation between the two taxa (see [25, 27]), which is a focus of our model.

In addition to general concerns for gene drives such as the development of resistance, the following ecological hypotheses proposed in the literature are investigated: transmission of the gene drive between two hybridising subspecies of *Anopheles gambiae* sensu lato (s.l.) by vertical gene transfer [25, 30, 31]; possible long range dispersal or long distance migration [21, 32]; and the nature of spatially and temporally varying carrying capacities driven by precipitation and land use factors [22, 27, 33]. This model is designed to provide scenario based testing of structural hypotheses that formalise the current state of knowledge for key gene drive and population life history parameters.

Each of these structural issues are described in the following subsections:

## Choice of construct

Our focus is on the spread and persistence of a gene drive system with near-neutral fitness that incorporates an unavoidable small reproductive payload cost of expression of the nuclease (see [8] and [16]) through a spatially and temporally dynamic population with differential gene flow across sub-Saharan Africa. This scenario thereby evaluates the behaviour of an idealised nearly fitness-neutral population replacement gene drive system at the continental scale. Our analysis focusses on three alleles: the wild–type, the genetic construct for a population replacement gene drive and a resistant allele. Together these form a minimal gene drive spatial model (see [16]). Further, we assume that the gene drive is activated in a single locus in each parent's genetic code as in [16]. However, we avoid modelling a gene drive "payload" of a nuclease or effector gene as done in their paper. That is, the nuclease and effector gene may be considered as the same unit, with the effector gene either absent or nearly fitness neutral. In practice, an effector would also be likely to exhibit a genetic load on the receiving organism (e.g. [16]). Therefore the model predictions are deliberately optimistic in terms of the magnitude of spread and persistence of the construct, and provide an indication of the spread of a population replacement drive for an idealised effector with negligible fitness cost.

## Taxonomic resolution

We model an intervention where the genetic construct has been introgressed into locally sourced, wild-type *An. gambiae* s.s. or *An. coluzzii* mosquitoes, and subsequently released back into this local population. Depending on geographic location, these subspecies of the *An. gambiae* s.l. complex may introgress with each other and potentially other subspecies such as *An. arabiensis* [25, 34, 35]. Studies at the scale of sub-Saharan Africa often do not discriminate between *An. gambiae* s.s. and *An. coluzzii*. For example, [36] combined these two subspecies when plotting species distribution maps due to lack of data; *An. arabiensis*, however, was plotted separately. Similarly, [21] argued that currently there is a lack of available data for parametrising alternative life history strategies of *An. gambiae* s.s. and *An. coluzzii*, and so did not discriminate between these subspecies in their process model. Indeed, we are currently unaware of any analysis of genetic vector control strategies at this continental scale, with explicit spatial and temporal dynamics, that discriminates between these two co-dominant malaria vectors.

We include alternative subspecies in our model because this leads to altered population dynamics via interspecific mating and density dependence effects [30]. Here we explore two different approaches to species assignment of first generation hybrids: 1) species assignment by maternal descent, and 2) equal proportions.

## Larval carrying capacity

To parameterise a spatial model that discriminates between *An. gambiae* s.s. and *An. coluzzii*, and their inter– and intra–specific density dependence at the larval stage, we require spatially

explicit carrying capacity information about each subspecies, which is anticipated to depend on environmental and social covariates. As observation data is relatively sparse at the subspecies level, we approach the problem in two parts. First, we model the larval carrying capacity of the two subspecies taken together using a functional form. Second, we use empirical relative abundance data to spatially model the relative carrying capacities between subspecies.

**Total abundance.**   The carrying capacity of *Anopheles* species in Africa is often expressed as a function of rainfall. For example, [33] found exponentially weighting the past 4 days of rainfall gave the best fit when modelling the abundance of *An. gambiae* s.s. and *An. arabiensis* in Nigeria, an approach subsequently adopted by [37]; whilst [38] used a 7 day moving average of rainfall to model the carrying capacity of the aquatic population of *An. gambiae* s.s., *An. coluzzii* and *An. arabiensis* in Mali.

More complex functional forms invoke additional parameters such as the location (and sometimes length or size) of perennial, intermittent, permanent or human-associated water bodies, as in the models developed by [39] and [40]. The most relevant approach for our purposes, however, is that of [21], who group our two proposed subspecies *An. gambiae* s.s. and *An. coluzzii* in a spatially explicit, individual-based model, across an area of West Africa that exhibits significant environmental variation. They predict local larval carrying capacity based on rainfall, as well as level of access to temporary and permanent water courses. We adapt their results for our model of total carrying capacity for the aggregate of *An. coluzzii* and *An. gambiae* s.s.

**Relative abundance.**   *An. coluzzii* has only relatively recently been described as its own subspecies [29] after its earlier description as a molecular form within *An. gambiae* s.s. [41, 42]. Despite this taxonomic difficulty, several papers have examined differences in habitat preference between *An. gambiae* s.s. and *An. coluzzii*. In particular, [26] note that larval predation and competition has led to selection for temporary freshwater habitats in *An. gambiae* s.s. and conversely permanent habitats for *An. coluzzii*. They suggest that this leads to humidity and/or rainfall clines in relative abundance. Other sources provide data suggesting this is the case for rainfall [43–46], and that a particular chromosomal arrangement in *An. coluzzii* performs well in low-rainfall environments [47]. These conclusions, however, tend to be based on relatively small-scale observations or experiments. Some information on relative abundance at larger scales is available [48–51], but very little modelling has been done to quantify these differences across sub-Saharan Africa. So far only occurrence information has been widely used [52]. In contrast, the relative abundance of *An. gambiae* s.s. in its former definition (including *An. coluzzii*) versus *An. arabiensis* has long since been modelled and estimated across sub-Saharan Africa [53].

A notable exception in this context is [27], who develop a logistic regression model using abundance data of each subspecies across western sub-Saharan Africa to predict relative probability of occurrence of the two subspecies. They use model selection to select a subset of relevant spatial, climatic and land cover variables in their predictions. However, despite acknowledging the likelihood of nonlinear effects of some variables, they use only linear predictors in a logistic regression. We use their work as a starting point to develop a flexible neural network model to incorporate nonlinear relationships, along with additional predictors and newly collated records of relative abundance (VectorBase: [54]) to extend our predictions to include the rest of sub-Saharan Africa.

## Dispersal

Anopheline mosquitoes have historically been categorised as being unlikely to migrate long distances (with mean dispersal distances typically less than 1 km, maximum distances typically

no greater than 5 km). Although longer range dispersal events are possible and have been linked to mosquito-borne disease outbreaks, short range dispersal is supposed to predominate life history strategies [55, 56]. A recent empirical study [32], however, provides evidence for wind-driven long-range dispersal of *An. gambiae* s.s. and *An. coluzzii* mosquitoes in large numbers. These mosquitoes remain capable of reproduction and pathogen transmission [57], and are estimated to regularly travel much further than even a rare long-range dispersal event could achieve. Moreover, it has been recently suggested that *An. gambiae* s.l. populations in areas of low human density may also facilitate migration, gene flow or both [58].

### Aestivation

Another source of controversy is aestivation, in which mosquitoes become dormant during dry conditions in which they would not otherwise survive. While not proven to occur widely on a population scale, it is a popular hypothesis for wet season reemergence of *An. coluzzii* in the Sahel [59, 60]. Modelling studies that address this problem include [38] and [21]. The latter simulation study notes that rare persistent water sources provide a competing explanation for persistence through the dry season, as does long distance migration. In this model, these latter proposed processes are accommodated by spatially and temporally varying carrying capacities (see Larval carrying capacity above) and dispersal behaviour (see Dispersal above); aestivation is not explicitly modelled.

### Spatial scope

The spatial scope for this analysis includes all countries within the African region as defined by the United Nations geoscheme that are within the range of *Anopheles gambiae* s.l. (United Nations regions are listed here: https://unstats.un.org/unsd/methodology/m49/overview/). The spatial scope includes the range of *Anopheles gambiae* s.l. on the African continent and also island countries or territories of the African region where *Anopheles gambiae* s.l. is present, such as Madagascar, Mauritius, Comoros and São Tomé and Príncipe. *Anopheles gambiae* s.l. is also found in Cabo Verde [61].

## Materials and methods

### Spatio-temporal mosquito demographic model

We represent each combination of age class (*a*), sex (*s*), genotype (*g*) and subspecies of mosquito (*m*) as a separate scalar field $X_{a,s,g,m}(t, \mathbf{x})$ in a Partial Differential Equation (PDE) model with time *t* and 2D location $\mathbf{x}$. Mosquitoes are assumed to not persist in ocean regions, and we set a zero-population Dirichlet condition in those regions. We allow, however, for the possibility that mosquitoes can advect across the ocean to neighbouring islands for a maximum duration of one day.

We represent the model numerically by spatially discretising the mosquito population of sub-Saharan Africa into 5 km × 5 km grid cells using the Africa Albers Equal Area Conic projection (ESRI:102022), and temporally discretising in 1 day timesteps, using fourth-order Runge-Kutta to integrate between timesteps. This timestep was determined experimentally to be both numerically stable and accurate. Combined with the spatial resolution, the model can be run within a reasonable time frame, while still being suitable for modelling the relevant biological processes, with a mosquito home range being roughly one cell in size. Within a timestep, we model demographic processes, followed by diffusion (see Diffusion below), followed by advection (see Wind advection below). While our model is deterministic, we allow extinction to occur by making a continuous model correction: at the end of the demographic

processes step in each timestep, any cell with less than one total mosquito in a subspecies is set to zero for all classes of that subspecies.

We define $N_{subspecies}$ subspecies, such that each subspecies is given an integer number from $m = 0$ to $m = N_{subspecies} - 1$: in our main results, $N_{subspecies}$ is set to 2 to represent *An. gambiae* s. s. and *An. coluzzii* as $m = 0$ and $1$ respectively. We also discretise age into $N_{age}$ age classes in our model: in our main results, we use $N_{age} = 2$ (one newborn and one adult class), but we describe the general model for $N_{age} \geq 1$ here, as different values of $N_{age}$ may be more relevant for different model applications. For $N_{age} \geq 2$ age classes, the constant transition rate $b$ results in an exponential distribution of maturation times between larval classes, and between the oldest larval class and adults. When $N_{age} = 2$, as in our model, some larvae will quickly mature, potentially making the population more resilient to intervals without rainfall. Although we focus on $N_{age} = 2$, S2 Fig shows that results for $N_{age} = 6$ closely match results for $N_{age} = 2$ in S1 Fig, with resistance spreading slightly more slowly over time.

Ages vary from the "newborn" larvae age class $a = 0$ (into which all individuals are born) through to adults at $a = N_{age} - 1$, with $[1, N_{age} - 2]$ representing intermediate larval stages where these exist ($N_{age} > 2$). We here describe the PDE separately for these three stage types. Note that in our model we only track those mosquito eggs that produce viable larvae, hence our newborn class consists of larvae instead of eggs. Our numerical model is written in Cython (the Python programming language with C extensions: [62]) and visualisations are performed in the R programming language [63].

**Adults.** The PDE governing each scalar field representing adult populations $X_{N_{age}-1,s,g,m}(t, \mathbf{x})$ for $N_{age} > 1$ is given by

$$\frac{\partial X_{N_{age}-1,s,g,m}}{\partial t} = -d_A X_{N_{age}-1,s,g,m} + b X_{N_{age}-2,s,g,m}$$

$$+\nabla \cdot (D\nabla X_{N_{age}-1,s,g,m} - A\mathbf{V}(t, \mathbf{x})X_{N_{age}-1,s,g,m})$$

where subscript $s$ denotes sex ($M$ or $F$), $g$ genotype, and $m$ mosquito subspecies ((1) *An. gambiae* s.s. and (2) *An. coluzzii*). We model three alleles: $w$ for wild–type, $c$ for construct (gene drive system), and $r$ for resistant. This results in a set of six potential genotypes $G = \{ww, wc, wr, cc, cr, rr\}$. The vector field $\mathbf{V}(t, \mathbf{x})$ represents the wind experienced across the spatial domain at a specified time $t$ and place $\mathbf{x}$. Note that our model makes some modifications to the advection process for biological reasons, specified below in Wind advection. Other parameters are given in Table 1.

Here $N_{age} - 1$ indicates the adult age bracket. If $N_{age} > 1$ then $N_{age} - 2$ is the eldest larvae age bracket; the special case of $N_{age} = 1$ is considered below (Newborns and $N_{age} - 1$ model). The first term on the right-hand side ($-d_A X_{N_{age}-1,s,g,m}$) describes the mortality of adults, while the second term ($b X_{N_{age}-2,s,g,m}$) describes aging from the eldest larvae. The final term represents diffusion ($\nabla D\nabla X_{N_{age}-1,s,g,m}$) and advection ($-\nabla A\mathbf{V}(t, \mathbf{x})X_{N_{age}-1,s,g,m}$), with $A$ being the probability of an adult mosquito being advected by wind.

**Larvae.** For $a \in [1, N_{age} - 2]$, the populations are governed by

$$\frac{\partial X_{a,s,g,m}}{\partial t} = -d_J X_{a,s,g,m} + b\left[X_{a-1,s,g,m} - X_{a,s,g,m}\right] \tag{1}$$

As above, the first term on the right-hand side ($-d_J X_{a,s,g,m}$) describes the mortality of this age-bracket of larvae, while the second ($b[X_{a-1,s,g,m} - X_{a,s,g,m}]$) describes aging to/from older and younger age brackets respectively. Note that for $N_{age} \leq 2$ there are no such intermediate larvae.

**Table 1. Parameter definitions and estimates for final model ($N_{age} = 2$).**

| Variable | Name | Description | Estimate | Source |
|---|---|---|---|---|
| **Demographics** | | | | |
| $d_A$ | Adult mortality | Daily probability of mortality | 0.1 d$^{-1}$ | [21] (or see [40], [39], [64], [37]) |
| $d_J$ | Juvenile (larval) mortality | Daily probability of mortality | 0.05 d$^{-1}$ | [21] (or see [40], [39], [64], [38]) |
| $b$ | Larval transition rate | Number of days from an egg being laid to when it emerges as a sexually mature adult (when $N_{age} = 2$) or reaches next larval stage ($N_{age} > 2$) | 0.1 d$^{-1}$ | [21] (or see [39], [64], [37]) |
| $D$ | Diffusion coefficient | Typical rate of spread of population from a point source | 900 m$^2$ d$^{-1}$ | [40] (or see [65]) |
| $\lambda$ | Larvae per female | Expected number of larvae per female per day (wild type mosquitoes) | 9 female$^{-1}$ d$^{-1}$ | [21] (or see [40], [39], [37]) |
| $k$ | Relative probability of mating between subspecies | The relative probability that a female has offspring with a male of a different subspecies to her own ($k < 1$) | 0.01 | [25] (or see [66]) |
| $\alpha_{ij}$ | Lotka-Volterra competition between subspecies | The relative effect on subspecies X of a member of a different subspecies Y taking up its resources (and thus larval carrying capacity), as compared to a conspecific | $\alpha_{11} = \alpha_{22} = 1, \alpha_{12} = \alpha_{21} = 0.4$ | [25] |
| **Genetics** | | | | |
| $k_c$ | Probability of cleavage | | 0.995 | [16] |
| $k_j$ | Probability of non-homologous repair | | 0.02 | [16] |
| $k_n$ | Probability nuclease gene lost during homing | | $10^{-4}$ | [16] |
| $h_n$ | Dominance coefficient for nuclease expression | | 0.5 | [16] |
| $s_n$ | Cost of nuclease expression | | 0.05 | [16] |
| **Larval carrying capacity** | | | | |
| $K_m(\mathbf{x})$ | Larval carrying capacity | | Details in text (Larval carrying capacity) and below parameters | |
| $\alpha_0(\mathbf{x})$ | Permanent larval site population | | 0 (no permanent sites) | [21] |
| $\alpha_1$ | Contribution to breeding from rainfall | | 200,000 | [21] |
| $\alpha_2$ | Larval sites associated with rivers and lakes | | 200,000 | [21] |
| $\phi$ | Rate of carrying capacity population increase with rainfall | | 0.03 per mm rain per week | [21] |
| $\kappa$ | Rate of carrying capacity population increase with water bodies | | 0.8 per km water | [21] |
| $\delta$ | Replenishment rate of intermittent water sites with rainfall | | 0.03 per mm rain per week | [21] |
| **Model details** | | | | |
| $X(0, \mathbf{x})$ | Initial condition | | Details in text (Initial conditions) | |
| $A\,\mathbf{V}(t, \mathbf{x})$ | Advection | | Details in text (Wind advection in Materials and Methods, Wind advection in Results) | [32] |
| $t$ | Time domain for integration | | 2005–2015 | |

**Newborns and $N_{age} = 1$ model.** The PDE governing $X_{0,s,g,m}(t, \mathbf{x})$ is given by

$$\frac{\partial X_{0,s,g,m}}{\partial t} = -d_J X_{0,s,g,m} - b X_{0,s,g,m} + B(s, g, m) \quad . \tag{2}$$

Where $N_{age} = 1$, there is no age structure and the full PDE takes the form

$$\frac{\partial X_{s,g,m}}{\partial t} = -d_A X_{s,g,m} + B(s, g, m)$$
$$+ \nabla \cdot (D\nabla X_{s,g,m} - A\mathbf{V}(t, \mathbf{x})X_{s,g,m}) \tag{3}$$

Note that for readability, we only use subscripts when referring to mosquito population classes (i.e. $X_{a,s,g,m}$) and refer to parameters which differ by population class as functions of the offspring classes. Note that we do not include parental classes (i.e. $g_M, g_F, m_M, m_F$) in the function names, also for brevity, and that these are only defined explicitly in the function definitions.

In Eq 2, the first term $(-d_J X_{0,s,g,m})$ describes mortality of newborns, while the second $(-b X_{0,s,g,m})$ describes aging into the youngest age-bracket of larvae (or adults for $N_{age} = 2$). The final term describes the birth of newborn larvae of a given sex $s$, genotype $g$ and subspecies $m$, given all possibilities of the mother's and father's genotype and subspecies ($g_M, g_F, m_M$ and $m_F$; described in more detail later). For brevity, we describe it as the product of functions describing the relevant biological mechanisms:

$$B(s, g, m) = \lambda \max \left( 0, \ 1 - \frac{C(m)}{K_m(\mathbf{x})} \right) \times$$

$$\sum_{\substack{g_M, g_F, \\ m_M, m_F}} J(m) \, O(s, g) \, R(g_F, g_M) \, X_{N_{age}-1, F, g_F, m_F} \tag{4}$$

where the baseline fecundity rate is $\lambda$, the expected number of larvae per clutch of eggs per female per day, assumed produced by a mating of wildtype mosquitoes. The $J$ and $O$ terms represent subspecies inheritance and genotype inheritance (including sex) respectively, and are described below. In our main results, we keep the relative fecundity function $R(g_M, g_F)$ constant at $R = 1$, but this can readily be varied to represent reduced fecundity due to inviability of a gene drive construct—see S3 Fig for an example.

The $\max \left( 0, \ 1 - \frac{C(m)}{K_m(\mathbf{x})} \right)$ term models the effect of $K_m(\mathbf{x})$, the (spatially varying) larval carrying capacity for subspecies $m$, on the fecundity of that subspecies using a logistic function. The inclusion of a max term is to keep the model biologically plausible: without it, $C(m) > K_m(\mathbf{x})$ would mean that negative newborns are produced. Here $C(m)$ is the competition that a newborn of subspecies $m$ experiences from all larval populations:

$$C(m) = \sum_{m'=0}^{N_{subspecies}-1} \alpha_{m,m'} \sum_{a=0}^{\max(0, N_{age}-2)} \sum_{s \in \{M,F\}} \sum_{g \in G} X_{a,s,g,m'} \tag{5}$$

where $\alpha_{m,m'}$ represents the effect of competition of subspecies $m$ on subspecies $m'$, and the max term is here used to ensure that where $N_{age} = 1$, the larval carrying capacity instead applies to the adult population.

The function $J(m)$ returns the probability that a male adult of genotype $g_M$ and subspecies $m_M$ successfully mates with a female adult of genotype $g_F$ and subspecies $m_F$ to produce newborns of subspecies $m$:

$$J(m) = \frac{H(m_M, m_F, m) \, W(g_M, m_M, m_F)}{\sum_{g' \in G} \sum_{m'=0}^{N_{subspecies}-1} W(g', m', m_F)}, \tag{6}$$

where the number of available males of a subspecies and genotype is

$$W(g_M, m_M, m_F) = S(m_M, m_F) G(g_M) X_{N_{age}-1, M, g_M, m_M} \quad .$$

The numerator is the relative number of expected matings between a male of subspecies $m_M$

(through $S$) and genotype $g_M$ (through $G$) and the given female, while the denominator normalises the probability.

The relative probability of mating based on subspecies $S(m_M, m_F) = 1$ if $m_M = m_F$ and $k$ otherwise. The relative fitness based on genotype is:

$$G(g) = \begin{cases} 1 & g \in \{ww, wr, rr\} \\ (1 - h_n s_n) & g \in \{wc, cr\} \\ (1 - s_n) & g = cc \end{cases} \quad (7)$$

where $h_n$ and $s_n$ are adapted from the [16] model (see Table 1).

We compare two different scenarios for subspecies inheritance $H(m_M, m_F, m)$; maternal inheritance and equal inheritance. For maternal inheritance, the proportion of offspring born of a mating between subspecies $H(m_M, m_F, m) = 1$ if $m_F = m$ and 0 if $m_F \neq m$ (mother is always the same subspecies as her offspring). For equal inheritance, the proportion of offspring born of a mating between subspecies $H(m_M, m_F, m) = 0.5$ if $m_M \neq m_F$ (parents are different subspecies, i.e. cross-species offspring are equally split between subspecies). In both scenarios, $H(m_M, m_F, m) = 1$ if $m_M = m_F = m$ (both parents and offspring same subspecies) and 0 if $m_M \neq m$ and $m_F \neq m$ (both parents different subspecies to offspring). Where there are more than two subspecies, we also need to specify that $H = 0$ when $m_M \neq m_F$, $m_F \neq m$ and $m \neq m_M$ (i.e. parents and offspring are all of different subspecies).

The function $O(s, g)$ gives the probability of sex $s$ and genotype $g$ for the offspring. This probability depends on the genotypes of the parents such that

$$O(s, g) = i(g_M, g_F, g) \; p(g_M, g_F, s) \; . \quad (8)$$

The first term $i(\cdot)$ describes the probability of an offspring inheriting genotype $g$ given parents of genotypes $g_M$ and $g_F$. We again adapt the [16] model to our target genotypes, including the effect of the gene drive construct, and using their parameter values (see Table 1). However, instead of assuming full random mixing of alleles as in their model, we directly model genotypes of each parent (see Table 2).

The second term $p(\cdot)$ describes the proportion of male offspring (and thus sex bias mechanisms). In the results presented in the main paper, we keep $p$ constant at $p = 0.5$, though this can be readily varied to model constructs that induce sex bias in viable offspring (see S4 Fig for an example). The sex ratio is kept constant between subspecies [67].

**Diffusion.** Diffusion is modelled by using the nearest-neighbour finite-difference approximation to the Laplacian. That is, with timestep size $\Delta t$ and cell length $\Delta x$, the fraction of diffusing mosquitoes removed from one cell is $4D\Delta t/(\Delta x)^2$, where $\Delta x$ is the cell-size. One quarter of

**Table 2. Possible values for each offspring $g'$ in inheritance table $i(g_M, g_F, g')$, with probability of occurrence given by the number in parentheses, where allele probabilities $w = 1/2 - k_c/2$, $c = 1/2 + k_c(1 - k_j)(1 - k_n)/2$ and $r = k_c(k_n + k_j(1 - k_n))/2$.** Table is symmetric, so cells marked with * have the same values as their transposes.

| | | Female | | | | | |
|---|---|---|---|---|---|---|---|
| | **Genotype** | **wc** | **ww** | **wr** | **cc** | **cr** | **rr** |
| Male | **wc** | **ww**$(w^2)$, **wc**$(2wc)$, **wr**$(2wr)$, **cc**$(c^2)$, **cr**$(2cr)$, **rr**$(r^2)$ | * | * | * | * | * |
| | **ww** | **ww**$(w)$, **wc**$(c)$, **wr**$(r)$ | **ww**$(1)$ | * | * | * | * |
| | **wr** | **ww**$\left(\frac{w}{2}\right)$, **wc**$\left(\frac{c}{2}\right)$, **wr**$\left(\frac{w+r}{2}\right)$, **cr**$\left(\frac{c}{2}\right)$, **rr**$\left(\frac{r}{2}\right)$ | **ww**$\left(\frac{1}{2}\right)$, **wr**$\left(\frac{1}{2}\right)$ | **ww**$\left(\frac{1}{4}\right)$, **wr**$\left(\frac{1}{2}\right)$, **rr**$\left(\frac{1}{4}\right)$ | * | * | * |
| | **cc** | **wc**$(w)$, **cc**$(c)$, **cr**$(r)$ | **wc**$(1)$ | **wc**$\left(\frac{1}{2}\right)$, **wr**$\left(\frac{1}{2}\right)$ | **cc**$(1)$ | * | * |
| | **cr** | **wc**$\left(\frac{w}{2}\right)$, **wr**$\left(\frac{w}{2}\right)$, **cc**$\left(\frac{c}{2}\right)$, **cr**$\left(\frac{c+r}{2}\right)$, **rr**$\left(\frac{r}{2}\right)$ | **wc**$\left(\frac{1}{2}\right)$, **wr**$\left(\frac{1}{2}\right)$ | **wc**$\left(\frac{1}{4}\right)$, **wr**$\left(\frac{1}{4}\right)$, **cr**$\left(\frac{1}{4}\right)$, **rr**$\left(\frac{1}{4}\right)$ | **cc**$\left(\frac{1}{2}\right)$, **cr**$\left(\frac{1}{2}\right)$ | **cc**$\left(\frac{1}{4}\right)$, **cr**$\left(\frac{1}{2}\right)$, **rr**$\left(\frac{1}{4}\right)$ | * |
| | **rr** | **wr**$(w)$, **cr**$(c)$, **rr**$(r)$ | **wr**$(1)$ | **wr**$\left(\frac{1}{2}\right)$, **rr**$\left(\frac{1}{2}\right)$ | **cr**$(1)$ | **cr**$\left(\frac{1}{2}\right)$, **rr**$\left(\frac{1}{2}\right)$ | **rr**$(1)$ |

this amount is added to each of the 4 neighbours. If the neighbours happen to be inactive (such as on the model boundary) those mosquitoes are assumed to die instantly.

**Wind advection.**   Wind advection is expected to occur over very short time periods (overnight, as mosquitoes are not believed to travel during daylight; see [32]) and potentially very large distances (hundreds of kilometres, passively carried by the wind with negligible resistance; see [32]). As such, we explicitly trace the trajectories of mosquitoes from each cell during each timestep (1 day), as advected by the wind vector field $\mathbf{V}(t, \mathbf{x})$. We use the Cross-Calibrated Multi-Platform (CCMP) Ocean Surface Wind Vector Analyses dataset [68] to define this vector field; the data is available at the required daily timesteps over the time period required. We interpolate their vector field, given at 0.25 degree intervals (approximately 28 km at the equator), to fit our grid. In their own modelling, [32] set up two different scenarios, where mosquitoes mosquitoes travel either 2 hours or 9 hours a night. We explore both scenarios here, and also a third scenario with no wind advection.

## Larval carrying capacity

**Total abundance.**   We apply the method of [21] to estimate larval carrying capacity for both species combined. Specifically, we use Eq 1 from their paper:

$$K(t, \mathbf{x}) = \quad \alpha_0(\mathbf{x}) + \alpha_1(1 - e^{-\phi r(t,\mathbf{x})})$$
$$+\alpha_2(1 - e^{-\kappa[W_p(\mathbf{x})+W_n(\mathbf{x})(1-e^{-\delta r(t,\mathbf{x})})]}). \tag{9}$$

with parameters as estimated in their paper using Markov Chain Monte Carlo simulation with population data (see Table 1). The model incorporates rainfall ($r(t, \mathbf{x})$ in mm per week), as well as the availability of nearby rivers and lakes; these may be either permanent or intermittent ($W_p(\mathbf{x})$ and $W_n(\mathbf{x})$ respectively; details in their paper). It also has scope for modelling permanent larval sites ($\alpha_0(\mathbf{x})$) but we set this to be zero for our model. Whereas [21] uses settlements as their sites for mosquito populations, we are calculating populations across a square grid, so we assume that each cell contains a settlement as modelled in their paper if and only if there is a human population present in the cell. An implicit assumption in this approach is that areas without human settlements yield zero carrying capacity. This assumption is explored for a sparsely populated region as follows. To calculate $K(t, \mathbf{x})$ at each cell at each required timestep using Eq 9, we estimate human presence or absence by using data produced by [69] and publicly available from the Humanitarian Data Exchange (HDX; https://data.humdata.org/), except in Sudan, South Sudan and Somalia where these data are not available. There we assume human presence in all cells, which as an extreme case would facilitate the spatial spread of populations by local dispersal or advection. The other extreme, assuming human absence, is explored in S7 Fig: it was found to have minimal effect outside of these countries, which take up a relatively small area near the edges of the species range.

We use inland water data from the Digital Chart of the World (DCW) as in their paper, and obtain monthly rainfall data from NASA's Land Data Assimilation System (https://ldas.gsfc.nasa.gov/fldas). We also set the mosquito population and carrying capacity to zero for each subspecies in locations outside their range as estimated by the Malaria Atlas Project (see [3], updated by the Malaria Atlas Project, accessed 25 May 2021 from https://malariaatlas.org/ and available for use under a Creative Commons Attribution 3.0 Unported License, https://creativecommons.org/licenses/by/3.0/legalcode).

**Relative abundance.**   Once we have a measure of the combined larval carrying capacity $K(t, \mathbf{x})$, we need to separate this for two subspecies $K_1(t, \mathbf{x})$ and $K_2(t, \mathbf{x})$, which requires knowledge of the relative abundance of the two subspecies at each site. We use available field data collated by [27] on the number of captures of both subspecies at various locations across Africa

to flexibly predict a spatially-varying but temporally-constant relative abundance. We call this relative abundance $K_r(\mathbf{x}) = K_1(t, \mathbf{x})/K(t, \mathbf{x})$, or the proportion of *An. gambiae* s.s. at a site.

We first attempt to replicate the results of the [27] logistic regression model by independently sourcing the predictors that they used in their final model: these are Latitude, Distance to Coast, Annual Mean Temperature, Mean Temperature of Wettest Quarter, Mean Elevation, Annual Normalized Vegetation Difference Index (NVDI) and Annual Variation in NVDI. We then apply a logistic regression model to the (slightly different) predictors, as they did. As well as providing an independent verification of the results in their paper, sourcing the predictors ourselves allows us to use more data sources across sub-Saharan Africa, allowing us to extrapolate and test different modelling approaches.

Using our independently sourced version of the predictors, we then apply a dense feed-forward neural network to flexibly model the relative abundance function $K_r(x)$ (see S2 Appendix for details), using the likelihood from a binomial statistical model as our loss function (equivalent to cross-entropy in the Results). We perform leave-one-out cross-validation (jackknifing) on each of these models, comparing the replicated logistic regression model and the original [27] model with the neural network model, to test whether there is an increase in performance by incorporating nonlinear effects on the same set of variables.

The [27] model uses only allopatric sites (those where only one subspecies was detected) to train their models, as did our replication model and initial neural network model. As we are interested specifically in subspecies overlap, we include sympatric sites (where both subspecies were detected), including three new sites [70, 71] publicly available from VectorBase [54]. We also add further predictors of potential interest to the described neural network model: mean annual precipitation (BIO12) and precipitation of wettest quarter (BIO16), which were also of interest to their model but excluded by their model selection process, along with salinity [25] which was discussed in detail by [27] but not included as a model predictor. We then apply forward selection to select the model variables in the final model (see S2 Appendix for details). To ensure convergence in the larger range of scenarios, we here use a 50:50 training-testing split on the data, but otherwise keep the same network topology and approach. Once this process is complete, we then have our final estimate of $K_r(\mathbf{x})$, which we can then apply to the previously calculated $K(t, \mathbf{x})$ to obtain results for both $K_1(t, \mathbf{x})$ and $K_2(t, \mathbf{x})$.

## Parameters

Where possible and reasonable, we take parameter estimates from literature for our model (see Table 1), while checking that multiple sources give similar results.

**Wind advection.**  We use estimates from [32] to indirectly estimate the probability of adult mosquitoes being advected by wind. They estimate that 6 million *An. coluzzii* mosquitoes cross a 100 km line perpendicular to the prevailing wind direction every year. Over the course of a 2 or 9 hour flight (the two night-time flight times tested in their Methods) their calculated trajectories give displacements of 3–69 km and 47–270 km respectively (means 38.6 and 154.1; 95% mean CIs 37–41 and 140–168). If we assume that each migrating individual completes just one 2 or 9 hour overnight flight each way and are only counted once, the mosquitoes that will cross the imaginary 100 km line will largely come from a rectangle bounded by this line on one side, and another 100 km line positioned either 38.6 or 154.1 km upwind on the other side. So the area over which mosquitoes will cross this line will be roughly 100 km × 38.6 km = 3, 860 km$^2$ or 100 km × 154.1 km = 15, 410 km$^2$, within which 6 million *An. coluzzii* migrating mosquitoes are estimated to cross. This gives us a density of $1.55 \times 10^{-3}$ or $3.89 \times 10^{-4}$ migrating mosquitoes per square metre per year: converting to relevant units gives us 106.39 and 26.65 mosquitoes per 5 km × 5 km cell per daily timestep. If we can then estimate the overall density of

mosquitoes at their capture sites, we can then estimate the daily migration rate. Using the total abundance model from [21], the average carrying capacity for larvae at coordinates 14°N, 6.7°W (located in between the capture sites in Mali) across the simulation period (2005–2015) is 41, 526. Given the other parameters $d_J = 0.05$ d$^{-1}$, $d_A = 0.1$ d$^{-1}$ and $b = 0.1$ d$^{-1}$, at equilibrium we would expect approximately the same number of adults as larvae at carrying capacity. This gives a daily migration rate of $106.39/41, 526 \approx 2.6 \times 10^{-3}$ (or 1 in 390) for 2 hour migration; and $26.65/41, 526 \approx 6.4 \times 10^{-4}$ (or 1 in 1558) for 9 hour migration.

**Initial conditions.** As we expect similar numbers of larvae and adults, we set the initial conditions of the model for each age $a$, sex $s$, genotype $g$ and mosquito subspecies $m$ to

$$X_{a,s,g,m}(0, \mathbf{x}) = \begin{cases} \frac{1}{2} K_m(\mathbf{x}) & \text{for } g = ww \\ \\ 0 & \text{for } g \neq ww \end{cases}$$

In cells where both subspecies exist, there will initially be competitive effects reducing numbers of one or both subspecies. In addition, advection and diffusion will affect the equilibrium. For these reasons, we run the model once from the initial condition described for the 11-year time period from 1 January 2005 to 31 December 2015 as a "burn-in" period, in order for the population to approximately reach a dynamic equilibrium that might be encountered in the wild.

Once the burn-in period is complete, we simultaneously introduce 10,000 male mosquitoes of each subspecies (*An. gambiae* s.s. and *An. coluzzii* that are heterozygous with the construct (genotype *wc*) in 15 separate locations, and run the model for another 11-year period. We choose heterozygous mosquitoes to introduce as these have less fitness cost than homozygous mosquitoes (see Eq 7), increasing the chance of spread. The first five of these are placed on islands off the African mainland at the nearest suitable location (see below) to assess the potential for incursion of the genetic construct onto the mainland. The island sites chosen as illustrative examples are:

1. the Bijagós islands (off Guinea-Bissau),

2. Bioko (off Cameroon),

3. Zanzibar (off Tanzania),

4. Comoros (off Mozambique), and

5. Madagascar.

The other ten sites (numbered 6–15) were chosen to be as evenly spaced as possible across the range of *An. gambiae* s.s. where at least 10,000 mosquitoes of either subspecies is available year-round (the algorithm is described in S1 Appendix).

## Scenario tests

As mentioned in Wind advection above, we use the two scenarios from [32] that mosquitoes are passively advected with the wind for either 2 hours or 9 hours a night. We also add a third scenario of no wind advection at all, to fully explore the effect of wind on mosquito movement. These three scenarios are then combined with the two scenarios of subspecies inheritance (maternal and fifty-fifty) described in Eq 6 to make six total scenarios modelled.

## Results

### Larval carrying capacity

**Relative abundance.** To compare the effectiveness of the modelling approaches on the jackknifed mosquito data from [27] and VectorBase [54, 70, 71], where the model probability

of the subspecies being *An. coluzzii* at site *i* of *N* sites is $p_i$, and the true probability is $s_i$ (either 0 or 1 depending on subspecies present), we use four measures:

- the actual misclassification rate or "error rate", where the subspecies at site *i* is predicted to be *An. coluzzii* if $p_i > 0.5$, otherwise *An. gambiae* s.s.;

- the cross-entropy, which is equivalent to the binomial statistical model used to fit the neural network model, calculated as
$$-\Sigma_i[s_i \log(p_i) + (1 - s_i) \log(1 - p_i)]$$

- the expected number of misclassifications or errors,
$$\Sigma_i \text{abs}(p_i - s_i)/N$$

- and the root mean square (RMS) error,
$$\sqrt{\sum_i (p_i - s_i)^2 / N}$$

Table 3 shows that our neural network model performs as well (error rate), slightly better (expected error rate and RMS error) or much better (cross-entropy) on all of the measures. Its much better performance on cross-entropy is likely due to the fact that it is trained specifically to minimise cross-entropy, which may not directly correlate with lower error rates—for example, compared to the other measures, cross-entropy will much more harshly penalise a model for assigning a very low probability to an event which then occurs in the testing data.

Fig 1 illustrates the forward selection process. The chosen variables are Mean Annual Temperature, Latitude, Elevation and Distance to Coast—after this point, even the best chosen variable added to the model only increases the mean validation loss.

Note that the NVDI Coefficient of Variation (CV) actually had the lowest mean validation loss in the second round, but Latitude was instead selected for four reasons:

- The two had mean validation losses that were statistically indistinguishable even after 500 model runs, based on the bootstrap 95% confidence intervals,

- Comparing the results of the forward selection process where either the NVDI CV or Latitude is chosen in the second round, the latter gives much better mean validation loss results in *subsequent* rounds, demonstrating that the forward selection process is sub-optimally choosing the NVDI CV,

- Data for the NVDI CV is much harder to source, and the variable is a less directly biologically relevant predictor than Latitude, and

- The NVDI CV becomes a much less desirable predictor after Latitude is selected, suggesting that it has little to contribute to the model that is unique to it, and not already contributed by Latitude.

**Table 3. Comparison of modelling approaches using four different measures of accuracy.** The "Original" model is that of [27], the "Replication" is our attempt to replicate their model with available data for predictors, and "NN" is our neural network model as described in this paper, but using their predictors. The models are compared with the dataset described in their paper [27] and from VectorBase [54, 70, 71]).

|  | Original | Replication | NN |
|---|---|---|---|
| Error rate | 0.1051 | 0.1018 | 0.1051 |
| Cross-entropy | 170.06 | 172.43 | 158.30 |
| Expected error rate | 0.1595 | 0.1610 | 0.1513 |
| RMS error | 0.2839 | 0.2839 | 0.2728 |

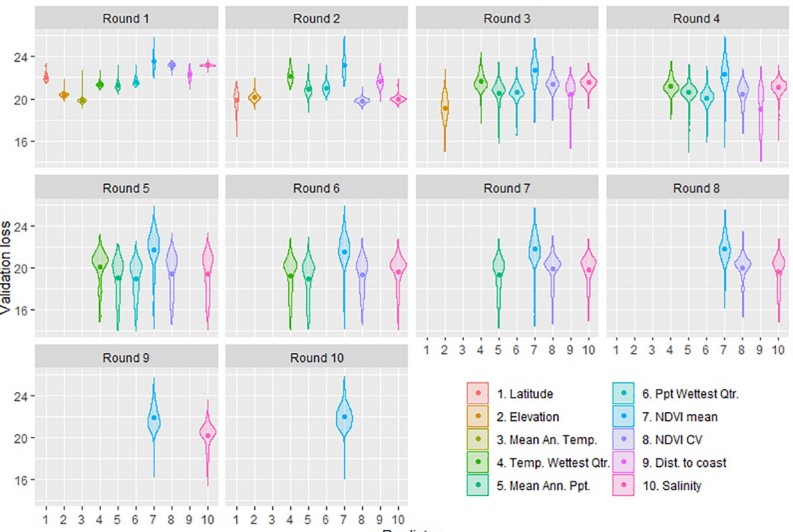

**Fig 1. The forward selection process for the neural network model that estimates the relative abundance of the two subspecies.** For each round of selection, the validation loss (a measure of how well the model predicts to novel data—lower is better) is shown with a different colour for each predictor. The width of the violin plots reflect the frequency of the validation loss across the 500 individual neural network runs, with the mean shown as a large dot. The forward selection process begins (Round 1) by calculating the validation loss for ten models that each include just a single different predictor. The predictor leading to the "best" model with the lowest mean validation loss is then accepted (Round 1: Mean Annual Temperature). The selection process continues (Round 2), calculating validation loss for nine models including Round 1's accepted predictor, and one new predictor. Similarly, the predictor leading to the best model is then accepted (Round 2: Latitude). This process is repeated until accepting a new predictor no longer improves the model. In our analysis, the selection process stopped after Round 4.

Fig 2 shows the final estimated relative abundance as a mean of 100 neural network model runs using the chosen variables, alongside the standard deviation of these runs. The relative abundance follows a similar pattern to that of [27], with *An. gambiae* s.s. mostly dominant except in some coastal areas and towards the Sahel. As expected, the variance between model

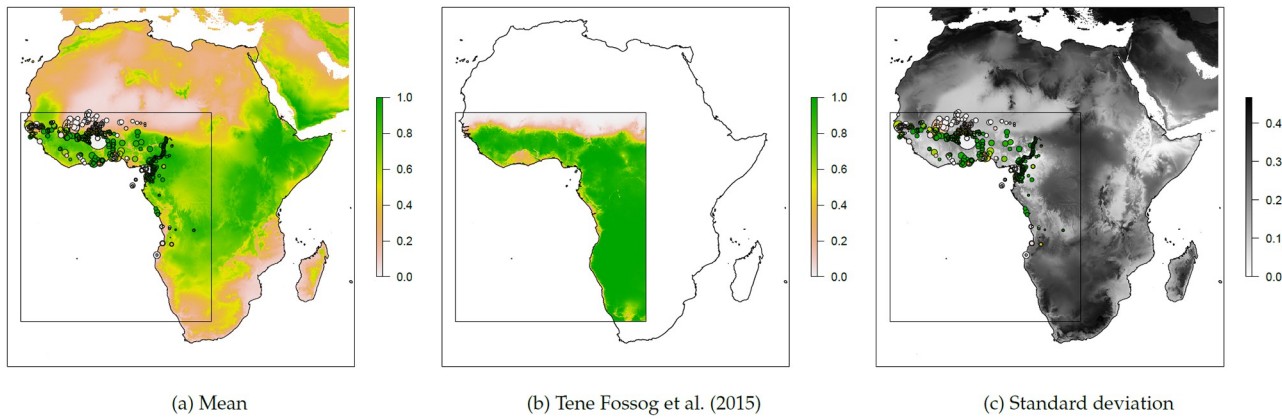

(a) Mean                    (b) Tene Fossog et al. (2015)                    (c) Standard deviation

**Fig 2. Summary statistics of relative abundance based on 100 neural network model runs.** Mean relative abundance (a) is given as the proportion of *An. gambiae* s.s. (as opposed to *An. coluzzii*) in the local mosquito population, with low proportions given as white and high proportions as green. Circles represent data points on which the model is trained, filled with colour representing the proportion measured at the given site. The corresponding results from [27] are given for the purposes of direct comparison (b). The standard deviation of relative abundance (c) between model runs is shown in greyscale, with white as low and black as high uncertainty of model estimates at a given site. The rectangle denotes the area of study used in [27]. Note that the relative abundance estimates cover areas where neither subspecies are expected to exist (see later figures). Base map from Natural Earth: https://www.naturalearthdata.com/downloads/10m-physical-vectors/10m-coastline/.

runs is mostly low around the data points in western sub-Saharan Africa upon which the model was trained, with much higher variance around central and southern Africa. Interestingly the model runs also yield low variance around Ethiopia in particular, where they predict *An. gambiae* s.s. to be dominant.

**Total abundance.** Fig 3 demonstrates the resulting estimated larval carrying capacity for each subspecies across the first year of modelling given the relative abundance model above and the [21] total abundance model, combined with information on human presence and the known distributions of each subspecies.

## Scenario tests

The choice of subspecies inheritance made no noticeable difference to the results for any of the three wind scenarios. Results shown use the maternal inheritance model.

Within the three wind scenarios, the zero-advection scenario experiences no noticeable mosquito transport on a continental scale, demonstrating that with the current selected parameters, advection is far more important than diffusion. We thus show figures for the 9-hour case below, contrast the 2-hour results in text, and present the full 2-hour and zero-advection results in Supporting Information.

Fig 4 shows the location and spread of the construct from the 15 introduction sites of the genetic construct (labelled 1–15) for the highest wind advection scenario (9 hours).

All of the islands were able to maintain a population with the genetic construct (Fig 5), with all except for Madagascar (Site 5) then invading the African mainland in subsequent years. There appears to be a barrier to dispersal in central Africa, with most introductions either remaining in western Africa (sites 1, 2, 7, 9, 14 and 15) or eastern Africa (sites 3, 4, 6, 8, 10–13). When advection was reduced to 2 hours, only the Bijagós (Site 1) and Zanzibar (Site 3) introductions were able to reach the mainland (see S6 Fig).

Fig 5 shows more detail about the introductions at each site. At all sites, the construct allele completely takes over from the original wildtype allele in a matter of months. Once the construct is established in the population, resistance builds slowly but surely: the heterozygous resistant genotype *cr* becomes noticeable after a couple of years, and is beginning to overtake the wildtype by the end of the 11-year simulation period, with the homozygous resistant genotype (*rr*) starting to become noticeable in the population. All sites show a similar pattern, despite differences in scale, subspecies composition and seasonality. Only *An. gambiae* s.s. persists in Sites 3, 4, 5, 7, 8, 11 and 12, and conversely only *An. coluzzii* persists in Sites 9 and 14. The two coexist in the remaining sites (1, 2, 6, 10, 13 and 15). All sites show a regular seasonal pattern to abundance based on rainfall (the only seasonal driver in the model, other than wind), with the possible exceptions of Sites 4, 6 and 8, which show slightly more variable seasonality patterns. Why this might be is unclear, though these sites are all in central to eastern Africa.

Full colour animations of all process model outputs are available in Supporting Information.

## Discussion

This study is the first continental scale model of population dynamics for two of the dominant malaria vector species in the *Anopheles gambiae* sensu lato species complex. The transient spread and persistence of a population replacement gene drive was predicted for the two hybridising subspecies *An. gambiae* sensu stricto and *An. coluzzii*. The two major factors that determine the spread of the gene drive at the continental scale were 1) potential wind-driven dispersal and 2) spatially and temporally varying carrying capacities for the two subspecies

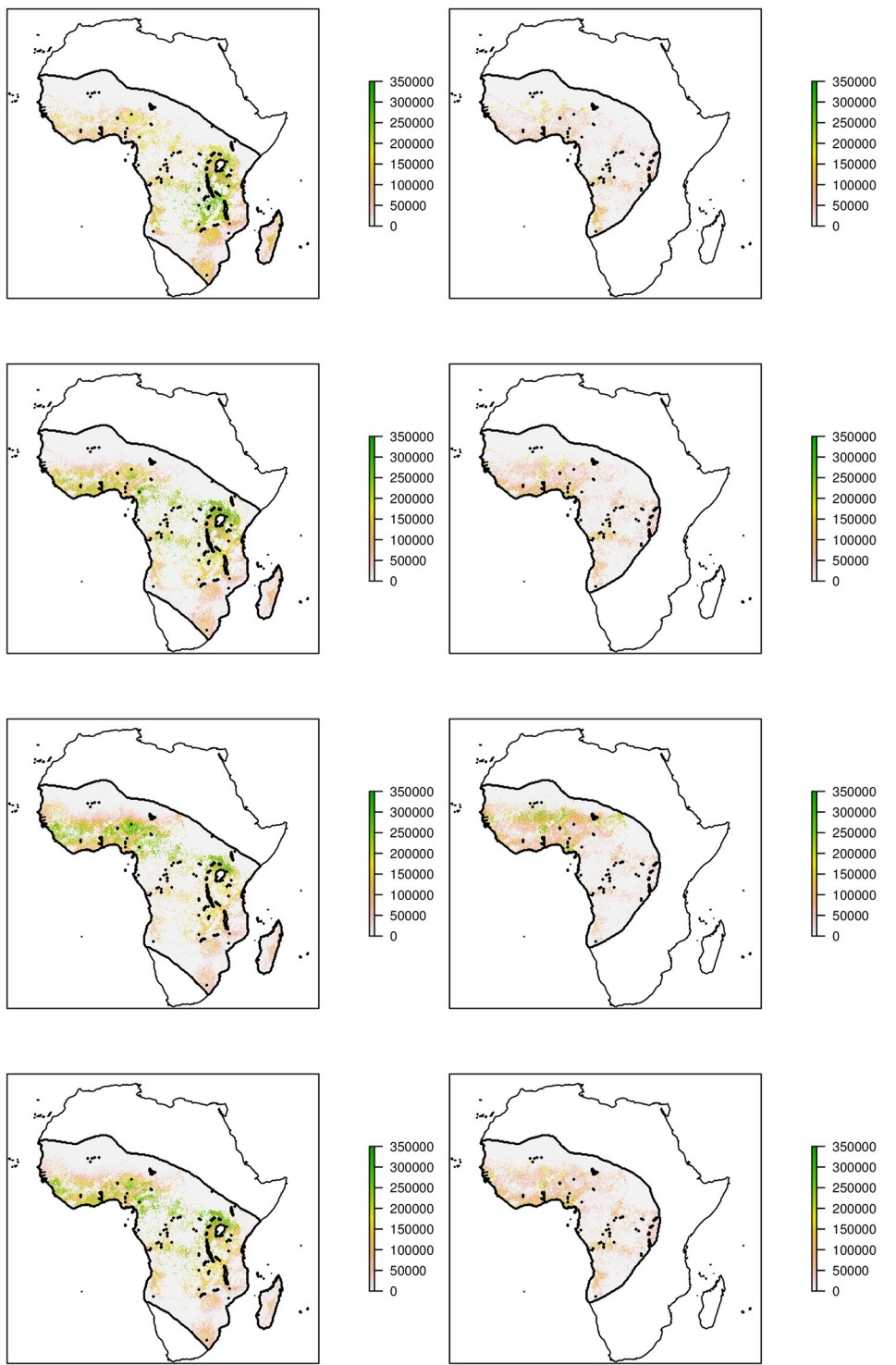

**Fig 3.** Estimated larval carrying capacity of *An. gambiae* s.s. (left) and *An. coluzzii* (right), for 2005 (the first year of modelling) in January (Southern Hemisphere summer), April (autumn), July (winter) and October (spring) from top to bottom. Base map from Natural Earth: https://www.naturalearthdata.com/downloads/10m-physical-vectors/10m-coastline/.

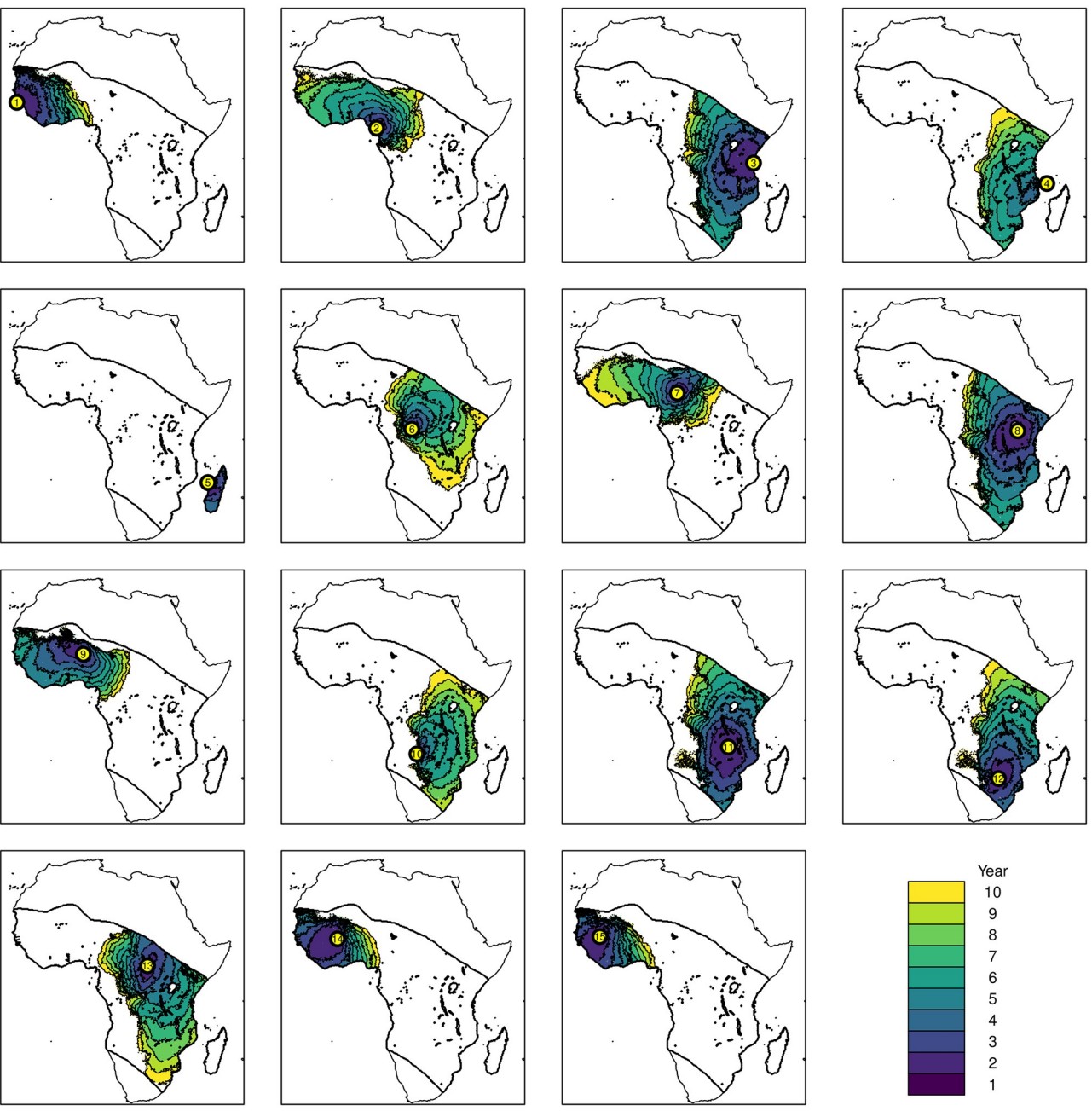

**Fig 4. The invasion front of the construct (defined as having at least two alleles, e.g. one *cc* or two *wc* mosquitoes, in a cell) from a selection of starting points, with a separate colour given for each year.** The island introductions are (1) the Bijagós islands (off Guinea-Bissau), (2) Bioko (off Cameroon), (3) Zanzibar (off Tanzania), (4) Comoros (off Mozambique) and (5) Madagascar. Base map from Natural Earth: https://www.naturalearthdata.com/downloads/10m-physical-vectors/10m-coastline/.

with both intraspecific and interspecific density dependence occurring at the larval stage. Our presented results are intended to demonstrate the plausibility of wide-scale spread of gene drive and structurally evaluate hypotheses of relevant life history strategies for these two dominant malaria vectors. The spatially explicit process model is designed to support more specific scenario-based assessments of both genetic and conventional vector control strategies.

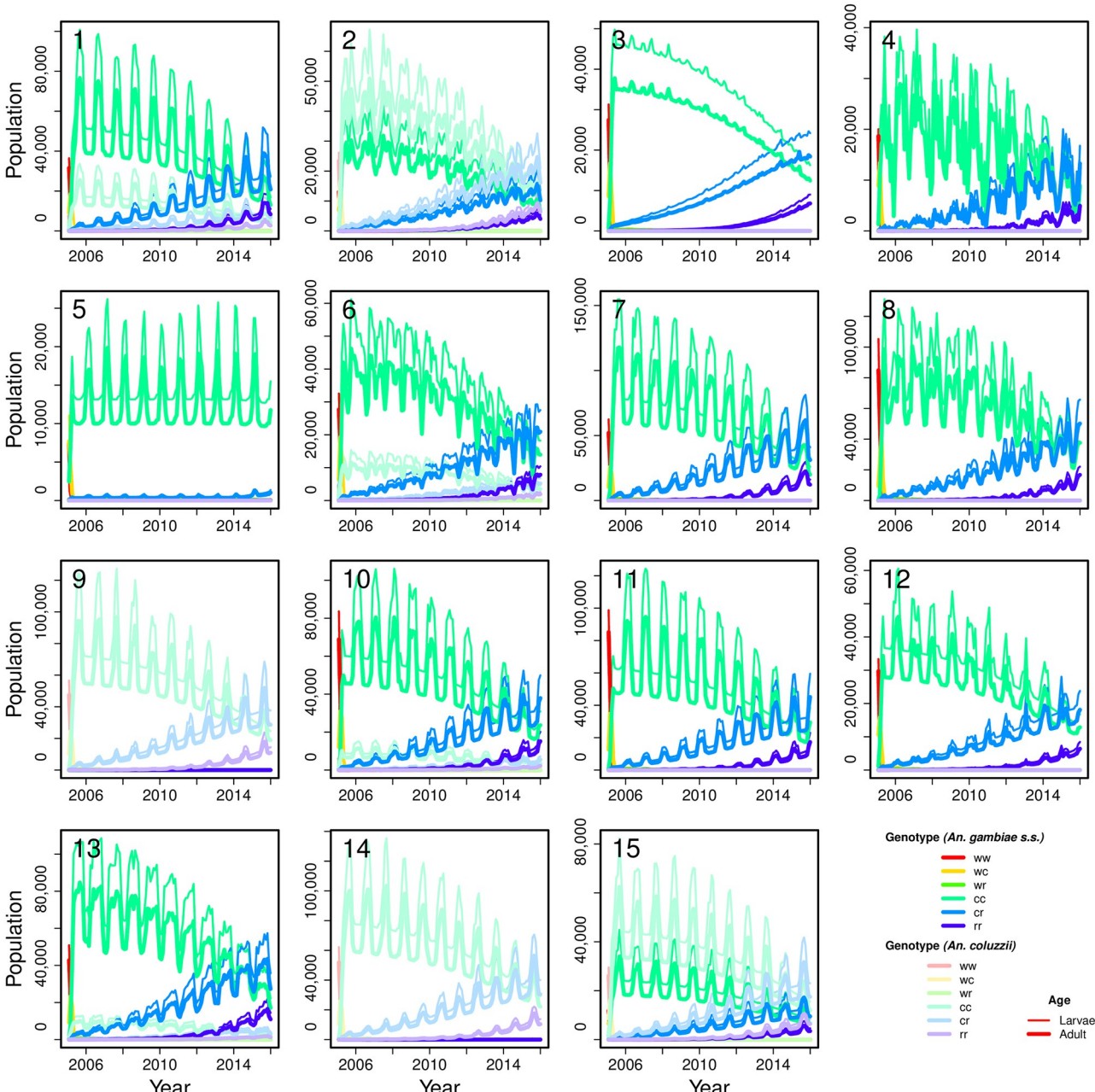

**Fig 5. The time series abundance of male mosquitoes at each introduction point (the sub-figure number corresponds to the release site; see Fig 4),** **separated by species, genotype and age (female mosquitoes occur in identical numbers to males in this model).** The colours correspond to genotype and the line thickness to age class.

Vertical gene transfer among the investigated subspecies is complicated by the spatially heterogeneous population structure, where introgression rates between *An. gambiae* s.s. and *An. coluzzii* vary with geographic location [25, 34, 35], and perhaps also over time [66]. In [25], they suggest that the rate of hybridisation between these subspecies depends on their relative frequency in the population. The model therefore tracks the relative number of available males by subspecies, and two alternative choices for species assignment of the first generation

hybrids was investigated. The model results were found to be robust to species assignment via maternal descent when compared to equal proportions of species among the first generation hybrids.

The home range of *An. gambiae* s.l. is typically believed to be less than 5 square kilometres, although there is evidence for long-range dispersal [32]. Spatial dispersal was modelled through two different mechanisms: a local diffusive process, which may be mediated by the presence of *An. gambiae* s.l. between human settlements [58], versus wind-driven advection. More generally, wind-driven dispersal events have been observed for mosquitoes including over ocean, although it is not understood whether or not wind-driven dispersal is a deliberate life history strategy for some species [56]. For genetic vector control strategies, the possibility of long range dispersal events are an important consideration that should be incorporated into the selection of field sites [12, 72]. Unsurprisingly our results indicate that passive wind-driven advection, if present, can greatly increase the speed and spatial footprint of the invasion front for a near-neutral population replacement gene drive (Fig 4, Scenario tests).

The importance of wind-driven advection as a dispersal strategy for *An. gambiae* s.l. will likely be a key uncertainty in future risk assessments for genetic control strategies. The large difference in the dispersal between the 9 hour (Fig 4) and 2 hour results (S5 and S6 Figs) have important implications for regional (trans-national) governance arrangements, the scope of stakeholder engagement activities and the degree of geographic containment that islands might provide during a staged-release strategy. In our simulated releases of the gene drive on islands, only Madagascar was sufficiently distant to prevent spread to the mainland under the 9 hour wind-assisted dispersal. Even when reducing wind-assisted dispersal to 2 hours, two of the five island releases resulted in spread to the mainland. Simulated introductions onto islands such as Bioko (32 km from the mainland) spread to the mainland, but only under the 9 hour scenario. Hence, only this result is consistent with recent genomic analysis that shows mosquito populations on this island are not isolated from the mainland [73]. Although our results make the potential scale of wind-mediated spread of gene drive clear, exactly how and where this spread occurs in the relevant mosquito taxa, and the mosquito behaviour that helps drive it, is still being studied [32, 74]. In addition, while data on previous wind patterns is readily available, predicting future wind patterns can only be done in very general terms, especially given the added complications and uncertainties caused by anthropogenic climate change. These issues together mean that future predictions of gene drive spread will likely involve high levels of uncertainty.

Environmental carrying capacity is another key factor that determines the spread and persistence of the simulated gene drive releases, as well as the wild-type population abundance. The introgression of the gene drive tended to initially follow seasonal patterns of carrying capacity driven by precipitation (Fig 5). Later, however, the importance of resistance gradually overcame the drive within 11 years despite the spatio-temporal variability of the population abundances and a relatively mild genetic load imposed by the construct. Resistance is a recognised challenge for gene drive systems (e.g. [13, 14, 75]) for which various counter-strategies have been proposed [16, 24, 76, 77]. Our simulations assume resistance alleles arise with frequency $r$ (Table 2) determined by the probability of cleavage ($k_c$), the probability of non-homologous repair ($k_j$) and the probability that the nuclease gene becomes non-functional due to mutation of the target site during homologous repair ($k_n$) [16]. Our model, however, does not account for pre-existing resistance in wild type populations caused by sequence variation in the target locus. Hence the (nonetheless rapid) progression of resistance in hybridising and spatially heterogeneous populations shown here could be underestimated, further emphasising the importance of managing drive resistance.

Our characterisation of environmental carrying capacities and abundance for the wild-type populations can accommodate alternative functional forms and parametrisation. Moreover, the functional forms and parametrisation of carrying capacity may be expected to change with time as climate [78, 79] and land use [80, 81] changes. The current lack of quantitative, species-specific, data on mortality and dispersal within the *An. gambiae* s.l. complex, however, limits our ability to parametrise relationships such as the larval carrying capacity (Eq 9) in a species-specific fashion [21], beyond a few, spatially limited, empirical studies (e.g. [38]). We anticipate that as data from entomological surveys, coupled with species differentiation through genetic methods (e.g. [34, 35, 54]) is increasingly centralised, then more detailed parametrisations will become possible at the subspecies level.

A lack of data also constrains our ability to compare alternative hypotheses of how *An. gambiae* s.l. populations persist in marginal habitat zones such as the Sahel, where observed patterns of seasonal abundance can be explained by either aestivation, permanence of larval microhabitats (i.e., non-zero carrying capacities during the dry season) or long range dispersal [21]. Although persistence in the Sahel can be explained without an explicit aestivation model (see Figs 3 and 5), aestivation may nevertheless be an important life history strategy for some species within the *An. gambiae* s.l. complex in the Sahel [60]. As for wind-driven dispersal, the relative importance of aestivation suffers from limited data.

The process model allows for the scenario-based testing of genetic and conventional vector control strategies. For genetic vector control, the sex-differentiated compartment model allows for both population replacement and population suppression gene drives. For the latter, the relative frequency of males and females may be an important component of the gene drive system [9] and also for sex-biased, self-limiting forms of genetic vector control strategies [4–6]; the model has the flexibility to accommodate sex bias from either maternal or paternal descent (Newborns and $N_{age}$ = 1 model above). Conventional control strategies that target adult females or larval stage mosquitoes can be accommodated by increasing mortality rates in locations and times dependent on the intervention scenario (sensu [38]), or by reducing carrying capacity for interventions where larval habitat is removed.

Vector control is a key component in strategies developed to combat mosquito-borne and vector borne diseases [82]. The deployment of vector control strategies, whether genetic or conventional, into hybridising spatially heterogeneous populations will require the development of spatially explicit models. These models should be constructed at a spatial and temporal scale that is commensurate with the intervention and include the possibility of resistance, which is not only a feature of genetic methods such as gene drive systems but is also an expected development for conventional strategies such as insecticide applications [24, 83]. Numerical simulation based scenario assessments can be used to investigate alternative ecological hypotheses in concert with proposed vector control intervention strategies. The spatio-temporal model developed here shows the importance of resistance, vertical gene transfer among hybridising subspecies, long range dispersal, spatio-temporal variability in larval mosquito habitats and deployment strategies at the continental scale.

## Supporting information

**S1 Fig. Closed population plot.** Time series plot of Site 6 (**as in** Fig 5) but with a closed population, i.e. no diffusion or advection. The colours correspond to genotype and the line thickness to age class.
(PDF)

**S2 Fig. Age classes plot.** Time series plot of Site 6 **as in** S1 Fig but with 6 age classes, keeping overall larval emergence period and mortality constant. This result is more closely analogous

to a fixed larval emergence time than to the exponentially-distributed version in the main model. The first five age classes combined are plotted here under "larvae". The colours correspond to genotype and the line thickness to age class.
(PDF)

**S3 Fig. Reduced fecundity plot.** Time series plot of Site 6 **as in** S1 Fig but with $R(g_M, g_F) = 0.05$ where $g_M \neq ww$ or $g_F \neq ww$. The colours correspond to genotype and the line thickness to age class.
(PDF)

**S4 Fig. Male sex bias plot.** Time series plot of Site 6 **as in** S1 Fig but with $p(g_M, g_F, s)$ modified so that any presence of the genetic construct in the father (*wc*, *cc* or *cr*) results in a 95% male sex bias in offspring. Dashed lines represent males, solid lines females. The colours correspond to genotype and the line thickness to age class.
(PDF)

**S5 Fig. 2 hours advection time series plot.** The time series abundance of male mosquitoes at each introduction point **as in** Fig 5, **but using 2 hours instead of 9 hours advection**, separated by species, genotype and age (female mosquitoes occur in identical numbers to males in this model). The colours correspond to genotype and the line thickness to age class.
(PDF)

**S6 Fig. 2 hours invasion front plot.** The invasion front of the construct at each introduction point **as in** Fig 4, **but using 2 hours instead of 9 hours advection**. A separate colour is given for each year. The island introductions are (1) the Bijagós islands (off Guinea-Bissau), (2) Bioko (off Cameroon), (3) Zanzibar (off Tanzania), (4) Comoros (off Mozambique) and (5) Madagascar. Base map from Natural Earth: https://www.naturalearthdata.com/downloads/10m-physical-vectors/10m-coastline/.
(PDF)

**S7 Fig. Human absence invasion front plot.** The invasion front of the construct as in Fig 4 for Site 6, but assuming human (and thus mosquito) absence in Sudan, South Sudan and Somalia (national boundaries given in red, as defined by the UN Office for the Coordination of Humanitarian Affairs). Base map from Natural Earth: https://www.naturalearthdata.com/downloads/10m-physical-vectors/10m-coastline/.
(PDF)

**S1 Video. 9 hours *Anopheles gambiae s.s.* animation.**
(GIF)

**S2 Video. 9 hours *Anopheles coluzzii* animation.**
(GIF)

**S3 Video. 2 hours *Anopheles gambiae s.s.* animation.**
(GIF)

**S4 Video. 2 hours *Anopheles coluzzii* animation.**
(GIF)

**S1 Appendix. Algorithm for placing introduction sites.**
(PDF)

**S2 Appendix. Neural network details.**
(PDF)

**S3 Appendix. Description of illustrative animations.**
(PDF)

## Acknowledgments

The authors wish to thank CSIRO Data61, CSIRO Health and Biosecurity and the Foundation for the National Institutes of Health for their ongoing support and encouragement. We would also like to thank Roslyn Hickson and Mathieu Legros for their thoughtful comments.

## Author Contributions

**Conceptualization:** Nicholas J. Beeton, Andrew Wilkins, Adrien Ickowicz, Keith R. Hayes, Geoffrey R. Hosack.

**Data curation:** Nicholas J. Beeton, Keith R. Hayes, Geoffrey R. Hosack.

**Investigation:** Nicholas J. Beeton, Andrew Wilkins, Geoffrey R. Hosack.

**Methodology:** Nicholas J. Beeton, Andrew Wilkins, Adrien Ickowicz, Keith R. Hayes, Geoffrey R. Hosack.

**Project administration:** Keith R. Hayes, Geoffrey R. Hosack.

**Software:** Nicholas J. Beeton, Andrew Wilkins.

**Supervision:** Keith R. Hayes, Geoffrey R. Hosack.

**Validation:** Nicholas J. Beeton.

**Visualization:** Nicholas J. Beeton.

**Writing – original draft:** Nicholas J. Beeton, Andrew Wilkins, Adrien Ickowicz, Keith R. Hayes, Geoffrey R. Hosack.

**Writing – review & editing:** Nicholas J. Beeton, Andrew Wilkins, Adrien Ickowicz, Keith R. Hayes, Geoffrey R. Hosack.

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
