## [Decision Letter · Decision Letter 0]

1 Dec 2021

Dear Dr Beeton,

Thank you very much for submitting your manuscript "Spatial modelling for population replacement of mosquito vectors at continental scale" for consideration at PLOS Computational Biology. As with all papers reviewed by the journal, your manuscript was reviewed by members of the editorial board and by several independent reviewers. The reviewers appreciated the attention to an important topic. Based on the reviews, we are likely to accept this manuscript for publication, providing that you modify the manuscript according to the review recommendations.

Sincerely,

Claudio José Struchiner, M.D., Sc.D.

Associate Editor

PLOS Computational Biology

Thomas Leitner

Deputy Editor

PLOS Computational Biology

[LINK]

Reviewer's Responses to Questions

**Comments to the Authors:**

Reviewer #1: This manuscript describes a model for the population dynamics of malaria vectors An. gambiae and An. coluzzii across the continent of Africa. To my knowledge this is the first attempt to a. incorporate both these cryptic species, which have some capacity to interbreed, in a single model and b. model the population dynamics of either species at a continental scale. It is an ambitious goal, yet one that is well motivated by the prospect of gene drive technologies being developed for these disease vectors. The challenge of modelling at this scale is twofold: first, building an appropriate model, and second – the greater challenge in my view - setting plausible parameters from the limited assortment of published empirical data. The authors modelling approach is to numerically solve a system of PDE equations set to a pan-Africa grid (resolution 5km). I find this deterministic approach to be sensible, and will potentially provide a baseline comparator for future studies that examine effects of stochasticity per se. The model description was generally clear, though some parts could have been even clearer (details below). For parameterisation, the authors used a separate model to map the relative abundances of the two species based on environmental predictors and observation data. This in itself is an ambitious task and it wasn’t entirely clear to me why the authors chose to do it rather than use the output of Tene Fossog et al (2015), that they cite as a comparator (more on this below). In addition the numerous demographic parameters are set from various sources (many of which are highly uncertain).

The model construction and parameterisation sections are necessarily quite substantial, and by comparison the results section feels quite brief. Not many scenarios are considered and there is not much sensitivity analysis to determine the most important parameters; for example I expect the results will be pretty sensitive to the parameters of genotype fitness and hybridisation. This left me wondering if the principle aim of the manuscript is to describe and parameterise a model that can be used for addressing future questions relating to mosquito gene drives, or to address the questions in hand. I think the paper is more successful if the first aim is the principle one and I’d suggest rewording to place greater emphasis on it – describing a model that can be applied to other questions will no doubt be valuable to ongoing discussions about mosquito gene drives. The specific results that are presented can serve as an example of the use of the model. Overall I am impressed with this manuscript, I think it will be a valuable contribution to gene drive modelling, though there are a number of aspects I think could be somewhat clearer.

1. Could you give more detail of how the diffusion aspect of the model works on the lattice – what is the numerical procedure? An illustrative diagram of this (and the model mechanics more generally) would be very helpful.

2. Relatedly, it wasn’t clear to me how the carrying capacity of a cell is modified by the presence/absence of human populations? Is it zero if there is no settlement and? And is there a difference if there is one or two or more settlements?

3. The relative abundance methods section was difficult to follow. Maybe you could start by stating the entomological data, and the environmental predictors you will apply? I was quite lost in the second paragraph “Using our version of the predictors…”, and I suspect others will be unless they are very familiar with these methods. I’d suggest replacing this section with a biology-focussed summary of what you did, and an appendix that spells out the details.

4. Similarly, the relative abundance results section is very technical and focusses on the statistics (details that could be in an appendix) rather than the biology. Fig 1 is not very clear and I’d suggest revising and moving to appendix. What I was hoping for was a figure showing how the results differ from those of Tene Fossog et al – could you re-work fig.2 to to show this? Comparison of the two models was a stated aim of the exercise, but it was not very present in the results or discussion.

Minor

1. Pg 6: “persistence of transgenic mosquitoes from release sites”.

2. Pg. 7: “ both spatially and temporally varying carrying capacities driven by precipitation and land use factors” is described as a “hypothesis”. I would describe it as a statement (a fairly obvious one at that). This paragraph needs re-wording.

3. Throughout ms – why do you refer to An. gambiae and An. coluzzii as “subspecies”, they are commonly agreed to be different “species”?

4. Pg. 12. transforming <1 mosquito → 0 mosquitoes is not “approximating stochastic effects” – it is a deterministic rule that allows extinction to occur in the model.

5. Table 1 misses out a lot of the model parameters – e.g. construct parameters and K parameters. Would be nice to have them all in 1 table, possibly with subsections.

6. Eqn 2.4 - Why don't you write J as a function of mM,mF,gM ?

7. Eqn 2.7. Can you explain the different fitness parameters : he,hn,se,sn. Why is the heterozygote cost quadratic?

8. Eqn. 2.9. alpha0 is a term for permanent larval habitat, not for aestivation.

9. how are the relative abundances implemented in the model equations? Is it via the alpha_ij terms?

Reviewer #2: This is an impressive model of an important topic -- the dynamics of a gene drive construct in two species of Anopheles mosquitoes across much of sub-Saharan Africa. This is the first such continent-scale model of gene drive mosquitoes, and the first large-scale model for population replacement. The introduction is clear, the methods and model well explained, the results well presented, and the discussion good, without over-interpretation. Some specific comments:

- Section 1.1 describes the gene drive system as 'neutral' and 'fitness-neutral, but it seems from 2.7 that there are fitness costs. This contradiction should be rectified. ('neutral' is also used in the discussion).

- The model is described as a PDE model with 1 day time steps. For the non-mathematician, it would be good to add a sentence for the reader how this differs from a difference equation model (if it does).

- In the results it is said that when addition is for 2 hours, only Zanzibar introductions reached the mainland, whereas in the discussion it is said that Bioko introductions also reach the mainland. This contradiction should be rectified.

- The model has only 3 alleles, which is fair enough, but perhaps it could be mentioned in the discussion that Beaghton et al. (2017) found that if one allowed mutations to disrupt the effector while leaving the driver intact, then eventually (far from the release site) one ends up with effector-less drivers, and in this sense the results presented in this paper may be optimistic. Any further thoughts on how the model might be extended would also be welcome.

**Have the authors made all data and (if applicable) computational code underlying the findings in their manuscript fully available?**

Reviewer #1: Yes

Reviewer #2: Yes

PLOS authors have the option to publish the peer review history of their article (what does this mean?). If published, this will include your full peer review and any attached files.

Reviewer #1: No

Reviewer #2: No

Figure Files:

Data Requirements:

Reproducibility:

References:

---

## [Decision Letter · Decision Letter 1]

12 Apr 2022

Dear Dr Beeton,

Thank you very much for submitting your manuscript "Spatial modelling for population replacement of mosquito vectors at continental scale" for consideration at PLOS Computational Biology. As with all papers reviewed by the journal, your manuscript was reviewed by members of the editorial board and by several independent reviewers. The reviewers appreciated the attention to an important topic. Based on the reviews, we are likely to accept this manuscript for publication, providing that you modify the manuscript according to the review recommendations.

Sincerely,

Claudio José Struchiner, M.D., Sc.D.

Associate Editor

PLOS Computational Biology

Thomas Leitner

Deputy Editor

PLOS Computational Biology

[LINK]

Reviewer's Responses to Questions

**Comments to the Authors:**

Reviewer #1: I thank the authors for their thoughtful responses to my previous comments which have improved the manuscript. I only have a few remaining comments/ suggestions that I would like them to consider.

Medium-sized comments

• Figure 1 is slightly clearer than before but still fairly confusing. It needs a more intuitive description of how to interpret and what it is telling us – starting with a more intuitive y-axis. ( To the uninitiated, it is not clear whether loss is good or bad..). The legend is inaccurate – there are no small dots/ labelled dots.

• Which of the 4 distance metrics (page 22/23) are the final results based on? all of them? In which case how are they combined?

• Figure 2. your map does not exclude areas where there are no gambaie or coluzzii mosquitoes, giving the impression that colluzzii is dominant in - South Africa, Madagascar, Spain (?). How do you account for this? Contrast with Fig 3 where you have species range boundaries, and blank outside them.

• Also Figure 2, I’m not sure how to interpret (c) - The standard deviation of relative abundance. Can you explain this in a more intuitive way – is it a measure of uncertainty in the prediction, or predicted variation in the presence of each species? The legend could be clearer on this.

• Will you make the predicted layer for relative abundance freely available to other researchers? This would be useful.

Minor comments/ suggestions

• In the author summary the wording “ ..eventually stopping due to the emergence of resistant alleles.” struck me as quite clumsy and not a good description of what happens. “the drive allele is eventually ousted by a resistant allele” or some such would be more accurate.

• Page 3 ‘North & Godfray 2018’ does not model a suppression drive – I think its the wrong reference (North et al 2020?).

• Page 7 “ information has been generally used” what does ‘generally’ mean here?

• Page 9. Surely m=N_subspecies -1 (you don’t have the -1)?

• Also p9. “the constant transition rate b results in an exponential distribution of maturation between larval classes”. More accurate to say here “maturation from larvae to adult”.

Reviewer #2: Responded very well to the previous comments. No further comments

**Have the authors made all data and (if applicable) computational code underlying the findings in their manuscript fully available?**

Reviewer #1: Yes

Reviewer #2: None

PLOS authors have the option to publish the peer review history of their article (what does this mean?). If published, this will include your full peer review and any attached files.

Reviewer #1: No

Reviewer #2: No

Figure Files:

Data Requirements:

Reproducibility:

References:

---

## [Editor Report · Decision Letter 2]

22 Apr 2022

Dear Dr Beeton,

We are pleased to inform you that your manuscript 'Spatial modelling for population replacement of mosquito vectors at continental scale' has been provisionally accepted for publication in PLOS Computational Biology.

Best regards,

Claudio José Struchiner, M.D., Sc.D.

Associate Editor

PLOS Computational Biology

Thomas Leitner

Deputy Editor

PLOS Computational Biology

---

## [Editor Report · Acceptance letter]

27 May 2022

PCOMPBIOL-D-21-01799R2 

Spatial modelling for population replacement of mosquito vectors at continental scale

Dear Dr Beeton,

I am pleased to inform you that your manuscript has been formally accepted for publication in PLOS Computational Biology. Your manuscript is now with our production department and you will be notified of the publication date in due course.

With kind regards,

Andrea Szabo
